# Treatment of Rheumatoid Arthritis with Gene Therapy Applications: Biosafety and Bioethical Considerations

**DOI:** 10.3390/biotech10030011

**Published:** 2021-06-23

**Authors:** Zinovia Tsitrouli, Maria-Anna Akritidou, Savvas Genitsaris, Gijsbert van Willigen

**Affiliations:** 1School of Humanities, Social Sciences and Economics, International Hellenic University, 57001 Thermi, Greece; ztsitrouli@ihu.edu.gr (Z.T.); makritidou@ihu.edu.gr (M.-A.A.); 2Section of Ecology and Taxonomy, School of Biology, National and Kapodistrian University of Athens, Zografou Campus, 16784 Athens, Greece; genitsar@biol.uoa.gr; 3Leiden University Medical Center, Department of Health, Safety and the Environment, Leiden University, 9500 Leiden, The Netherlands

**Keywords:** rheumatoid arthritis, gene therapy, medical biosafety, environmental biosafety, adeno-associated virus, vector

## Abstract

Rheumatoid Arthritis (RA) is an autoimmune and inflammatory disease that affects the synovium (lining that surrounds the joints), causing the immune system to attack its own healthy tissues. Treatment options, to the current day, have serious limitations and merely offer short-term alleviation to the pain. Using a theoretical exercise based on literature, a new potentially viable therapy has been proposed. The new therapy focusses on a long-term treatment of RA based on gene therapy, which is only active when inflammation of the joint occurs. This treatment will prevent side effects of systemic application of drugs. Furthermore, the benefits of this treatment for the patient from a socio-economic perspective has been discussed, focusing on the quality of life of the patent and lower costs for the society.

## 1. Introduction

Rheumatoid arthritis (RA) is a long-term inflammatory and autoimmune disease that affects the synovium (lining that surrounds the joints), causing the immune system to attack its own healthy tissues. The process starts with the release of pro-inflammatory cytokines, especially tumor necrosis factor-α (TNFα) and Interleukin (IL-6 and IL-1), followed by the production of inflammatory cytokines in the joint (TNFα, IL-6, -15, -16, -17, -18, Interferon-γ (IFN-γ)). RA starts with painful swelling, which can lead, ultimately, to bone erosion and joint deformity [1]. Symptoms appear in smaller joints first (mainly in those that attach the fingers to the hands and the toes to the feet); as the disease progresses, symptoms tend to spread to bigger joints as well. In the plethora of cases, RA symptoms occur in the same joints on both sides of the body; a great number of patients with RA also experience symptoms that do not involve the joints, such as weight loss, fatigue, and weakness. It is not known why the immune system attacks healthy body tissue in RA, although a genetic component appears likely [2] and can increase the susceptibility to environmental factors that may trigger the disease.

Despite the improved understanding of RA pathophysiology over the past 20 years and the appearance of improved treatment options, severe RA can still cause physical disabilities, while therapy with most antirheumatic drugs, such as non-steroidal anti-inflammatory drugs (NSAIDS) and disease-modifying anti-rheumatic drugs (DMARDs) is palliative [3], alleviating inflammation but leaving the disease incurable, with some patients partially or not at all responding, short-term effectiveness [4], and unwanted associated systemic complications of immunosuppression [5]. Biological-based approaches have appeared as the most promising, using mainly monoclonal antibodies, recombinant forms of natural inhibitors, recombinant soluble TNF receptors, or anti-inflammatory cytokines, counteracting the released cytokines produced in the joint [3]. However, these therapies have serious limitations, such as high expenses, side-effects (i.e., nausea, low blood pressure, skin reactions, trouble breathing), and the requirement for repeated systemic injections [6].

The aim of this paper was to outline the steps that could lead to a successful gene therapy which would tackle the abovementioned limitations. Furthermore, potential biosafety concerns that may be linked to the proposed treatment have been identified and discussed. Furthermore, ethical dilemmas that could arise when administering the proposed therapy have been pinpointed.

## 2. A Potentially Viable Proposal

To overcome the limitations and difficulties of the present treatments, genetic therapies for RA offer the possibility of delivery of the therapeutic gene product to the disease site and, thus, prevent side effects by systemic injections or infusion, while enhancing efficacy and achieving local long-term expression, with endogenous production of high concentrations of the therapeutic agent. The overall goal for the treatment of patients with RA should not merely be alleviating the pain, but also achieving remission or at least low disease activity for all patients and preventing irreversible damage to the diseased joints. Since most, if not all, of the forms of RA result in the inflammation of the joint, and thus, share the process of inflammation, a gene therapy approach for RA, aiming either at inhibiting proinflammatory cytokines and/or overexpressing anti-inflammatory cytokines [7], could be promising. In this context, and given the fact that the overproduction of inflammatory cytokines by fibroblast-like synoviocytes (FLSs) is believed to play a pivotal role in the development and progression of RA [8], we have proposed a therapy that would overall suppress inflammation, by expressing anti-inflammatory cytokines (see Figure 1 for a schematic representation).

Regarding the vectors of choice, the ideal vector should transfer a precise amount of genetic material into each target cell expressing the gene material, without causing toxicity. As a delivery method for the therapeutic gene, there are several choices available. The most obvious methods are plasmids carrying the therapeutic gene or viral vectors. Because a long time expression of the transgene is needed for treatment of RA, plasmid vectors are not an option, because they are known for only a short-term expression and often only suboptimal expression of the transgene, although there have been improvements made to overcome these difficulties [9]. Therefore, only viral vectors can be used to transfer the transgene. Viral vectors that will integrate into the genome or stay as an endosomal plasmid present in the cell have a preference. This limits the choice of vectors to viral vectors, such as retro- and lentiviral vectors and AAV [10]. Because the retro- and lentiviral vectors are known for insertional mutagenesis [11], the preferred vector is AAV. In the absence of a helper virus or genotoxic factors, AAV DNA can either integrate into the host genome at a predefined spot (chromosome 19) or persist in an episomal form [12]. This makes AAV the vector of choice, because it fulfils all the criteria needed for an effective therapy for RA.

Adeno-associated virus (AAV) is preferred, because it is safe, effective, and less immunogenic than other vectors. Genetic modifications of human cells can be done either by an ex vivo or in vivo approach. Both methods are possible in RA treatment and have been used in different studies [13]. The fact that modified cells were cleared shortly after intra-articular injection was the main disadvantage in several ex vivo studies [14], thus making in vivo delivery a preferable approach for RA treatment. AAV is commonly used in in vivo studies where the goal is long-term expression, as in RA, because this lowers the frequency of treatment administrations [15]. Specifically, for in vivo gene delivery to the joint by direct intra-articular injection, AAV is safer than other unsuitable-for-clinical-translation vectors that are inflammatory, immunogenic [14], and can provide more extended periods of transgene expression than non-viral vectors [16].

When it comes to the promoter, a promotor of the pro-inflammatory gene that is active during the onset of an inflammatory response in the joint is preferred, since in this way, expression of the therapeutic gene can be achieved locally and specifically when RA-related inflammation arises [17]. For this purpose, promotors of TNFα, IL-1α, Cyclooxygenase-2 (Cox2), or nuclear factor kappa-light-chain-enhancer of activated B cells (NF-kB) would all be suitable to regulate expression of the therapeutic gene, as they are upregulated during inflammation. Finally, the therapeutic gene needs to be an anti-inflammatory agent that will alleviate the phenomenon of inflammation in the joints. Τhere are numerous choices, but IL-4 [18] and IFN-β [19] are among the best candidates due to their anti-inflammatory functions.

## 3. Biosafety Considerations

Using viral vector systems for gene therapy as treatment options for several diseases is promising, but viral vector delivery remains risky and is still under study to ensure safety and efficacy during clinical trials. The safety of a gene therapeutic agent can be viewed from different angles. First is the risk for the laboratory worker and medical staff, second is the risk from a medical point of view, i.e., risk for the patient, and third is the risk for the environment. This third category also includes the risk for the patients’ offspring. However, and especially for AAV, the vector of choice in our case, safety concerns are limited, since AAV does not cause any known disease [20]. Furthermore, the risk for the laboratory worker and the medical staff will be negligible when standard hospital hygienic measures are in place. These will prevent contact with the AAV-particles during normal handling and during incidents. Most concerns are related to the preexisting immunity to human AAV vectors and the related integration into the host genome, which, if it happens at all, is random and could lead to accidental activation or inhibition of endogenous gene expression [21]. In this sense, medical and environmental risks are not related strictly to AAV and are considered, as already mentioned, rather safe, but mostly in relation to other parameters of the approach.

### 3.1. Biosafety for Lab and Medical Staff

In terms of laboratory precautions, AAVs are classified as Risk Group 1 [22]. Viral manipulation should be performed in a Biosafety Laboratory 1, with adequate biohazard signs, while manipulation in the same Biosafety Cabinet with other materials must be avoided to prevent contamination of the gene therapeutic agents. As already mentioned, the risk of an AAV vector for lab and medical staff is negligible. Health employees work using the standard hospital hygiene measures. These measures would prevent any direct contact with patient material, even if shedding occurred. During the injection of the AAV vector into the joint, the medical staff should wear personal protective equipment to prevent any exposure to the gene therapeutic agent. Furthermore, for the people working in the diagnostics labs, the risk of working with materials of the patient injected with the AAV is negligible. Normal working procedures in diagnostic labs are already sufficient to prevent unwanted exposure to AAV, even if shedding were to occur. The largest risk is during preparation of the syringe for injecting the AAV gene therapeutics. This procedure should be performed in a Biological Safety Cabinet Class 2 for sterile preparation, preventing unwanted exposure of the worker. In case of spills, sodium hypochlorite or quaternary ammonium compound are the recommended disinfectants, while alcohol is not an effective disinfectant against non- enveloped viruses, such as AAV [23]. Infection materials should also be decontaminated prior to disposal, generally using an autoclave, at 121 °C for 30–45 min [24].

### 3.2. Medical Risks

Before starting the clinical study, one of the very first questions that arises is which should be the joint in which the intra-articular injections will start. Since up to 75% of RA patients experience symptoms in the wrist [25], someone could suggest that this should be the joint of choice for gene therapy trials. However, studies have identified that injections into the wrist joint could result in complications [26]. Risk of septic arthritis following the injection of bacteria from the skin’s surface can enter the joint directly with insertion of the needle, while the synovium has little ability to protect itself from infection. Misplaced injections could potentially cause tendon rupture or even, in rare cases, nerve damage [27]. An infection of, or adverse events in, the synovial tissue in the wrist is hard to treat. The synovial tissue cannot be removed without causing any damage to the joint. When the wrist joint is damaged, the only option is to fixate it in an immobile position, which will hamper the function of the wrist and the mobility of the person. Replacement of the wrist joint while keeping the function of the joint is impossible. Because of this, we propose that the wrist joint is not the best option for starting gene therapy. Another option is the metacarpophalangeal joint (MCP) or knuckle. This is a small joint and one of the first joints affected by RA. From this joint, the synovial tissue can be easily removed, and if the joint is damaged, it can be easily replaced by an implant. This joint replacement would not affect the joint function. Thus, from a medical point of view, the MCP, as a joint for testing gene therapy, would be the joint of choice, because serious adverse events in the joint do not result in loss of function of the joint.

One of the potential benefits of gene therapy is that the therapy would be long lasting, and no repeated injections or oral medication would be needed. This decreases the burden for the patient (see also below in the “Bioethical considerations” section). For AAV, it has been shown that the expression of the transgene can be long lasting in different tissues. As already mentioned before, AAV is not only present in the episomal in target cells, but it also integrates into the genome. This integration gives rise to the long-lasting expression of the transgene. Studies have revealed a transgene expression using AAV vectors that lasts up to 10 years [28], making repeated injection unnecessary. Furthermore, the episomal AAV was shown to exist over a long period of time, with the expression of the transgene lasting up to six months in the liver [29]. For the first injection of AAV, a screen for pre-existing immunity can be performed. However, if repeated injections are necessary, an immune response against the therapeutic agent can be an issue. Several studies have already shown that suppression of the immune response can be successful when repeated injections are necessary [29]. Based on this, a gene therapy based on AAV would prevent daily medication, an additional burden of the RA patient.

### 3.3. Environmental Risks

AAV vector genomes remain episomal in target cells and are highly unlikely to integrate. Shedding from the host could only happen in rare cases, when the AAV integrates into the host cell chromosome, if both the adenovirus (or some other helper virus) and wild-type AAV are present. When it comes to the survival of this virus on surfaces, in the case of potential spills, sodium hypochlorite or a quaternary ammonium compound could be used to disinfect the area, since they are the recommended disinfectants against AAV [23]. Specifically concerning the animals used during the clinical trials of the proposed therapy and the potential risk caused by AAV, we should mention that, in some animal models, the integration of recombinant AAV has been associated with an increased incidence of tumor formation. However, this association has not been observed to occur in humans [30]. AAV vectors can shed from the patient into the environment, but also to the gonads. Both shedding events could give rise to unwanted effects of the treatment. Shedding to the environment can give rise to unwanted contact to the AAV particles of non-patient humans. Shedding to the gonads can result in germline transmission of the transgene. As already mentioned, a joint is closed by the synovial tissue that keeps fluid in the joint. When injecting the AAV vector into the joint, the synovial tissue would also protect the human body from the injected AAV vector. If injected correctly, no shedding from the joint would be possible. In case of damage of the synovial tissue, however, there will be shedding from the joint. Due to this, the AAV particles can become systemic. The biggest risk is the transduction of gonadal cells and the subsequent risk of germ line transmission. However, in studies where AAV was injected directly into the male gonads, no transduction of sperm cells was observed. The AAV preferred other cells in the gonadal tissue, such as the Leydig cells [31]. Long-term transduction of sperm cell-producing tissue was also not observed, and after a few cycles of sperm production, AAV in sperm cells was not detected [32]. Other gonadal tissues, not involved in spermatogenesis, could be positive for AAV over a longer period [32]. From this, it can be concluded that shedding has only a minor risk for germ line transmission and can be easily prevented.

## 4. Bioethical Considerations

Ethical questions arising generally in gene therapy, and specifically in our case, are not new to the debate, yet they are fundamental. Regarding the administration of the treatment, ethical concerns are of relevance, especially when it comes to the specific joint which should be chosen. Bearing in mind the complications that could result from a potential administration to the wrist, already mentioned under Medical Risks, we argue against such an option, due to the nature of the joint and the difficulty of treatment, in case of potential damage during wrong administration. We would opt for other joints, where this risk is rather limited and serious adverse events in these joints would not result in loss of function.

Rheumatoid arthritis, as mentioned above, can affect patients from different ages, but the disease usually has later-in-the-life onset symptoms, which mostly appear after the age of 35–50 [33]. This means that potential volunteers will, in the plethora of cases, belong in the middle-age and above age group. With most of them already having received other therapies (which most probably have failed), this could also mean that their symptoms are not light anymore. The first question that should be answered is how we will make our choice of volunteers. Should we choose people that have already received (inadequate) therapy or others, at earlier stages of the disease, with no prior experience with treatments? Especially given that, according to several published studies, older RA patients, at later stages of the disease, most probably receive less aggressive treatments than younger RA patients, even though they experience the same or more severe symptoms [34].

In the same context, we should not ignore questions regarding informed consent and its specific content, especially in cases of juvenile arthritis, where minors are not able to consent themselves. In our proposed treatment, risk is rather low, since AAV is a rather safe vector, which cannot have detrimental health effects, and in any case, the benefits from therapy outweigh the potential risk. However, as it happens generally with informed consent in minors, the rule should be that, besides their guardian’s or representative’s consent, their opinion must also be taken into consideration. It is important to opt for earlier intervention, given the severe complications and pain that come as a result when the disease progresses; thus, an early intervention would be more beneficial, rather than starting treatment when minors would have reached the legal age of consent. Taking into consideration the above, patients in each stage of the disease should participate equally in the study, since there is no just way in which we can weigh the costs and benefits between different stages and the respective level of pain, which should be avoided at all costs.

Moreover, RA is known to affect women more than men [35], and the question that subsequently arises is how this fact can potentially affect our chosen group of volunteers. It is probable that the percentage of women participating in the gene therapy trial will be bigger, since women suffer from RA in a higher ratio. However, can we say that, in the name of equality among patients, we would opt for including men and women in a ratio 1:1, or would such an option not serve equality among patients, since it would take into consideration criteria not directly connected to the level of pain and the severity of the symptoms? This difficult question correlates also with the criteria that would be used for inclusion/exclusion of the patients to ensure fairness in the selection procedure. We should not forget, at this point, socioeconomic parameters. It is true that people with higher economic feasibility would be informed easier, would more easily afford the related costs, and they would, thus, more easily participate in the trial.

Finally, in the case that treatment fails, and pain persists, there would be more dilemmas arising. More choices would have to be made in such a case, with regards to who would receive treatment: those that previously received it, but it failed, or those that are new in the trial? The same dilemma could arise in the case when patients have been treated on one side, but the joint in the opposite side also starts to present RA symptoms. Equality and justice among patients should be the main principles guiding our approach in all the aforementioned different situations, but the severity of the pain and the stage of the disease should play the most important role in our final decision.

## 5. Conclusions

Gene therapy can be a viable alternative to treat Rheumatoid Arthritis, a long-term inflammatory disease, alleviate the patients’ pain, and tackle the limitations of current treatments. The course of action we proposed here comes with biosafety concerns and bioethical dilemmas, which, should they arise, should be addressed with systematic approaches and guidelines. In particular, lab and medical stuff biosafety risks could be managed with the normal laboratory precautions, medical risks for the patient could be avoided if the suitable joint is chosen for the administration of the treatment, and environmental risks were not considered a point of concern in our proposed treatment, due to the characteristics of our vector of choice and the suggested solutions. Finally, the main ethical dilemmas to be considered included the choice of the joint for administrating the treatment, the choice of volunteers for the clinical trials, and the options of the patient, in case treatment fails. Equality among patients should guide the course of action in all the different situations that may accrue, but the severity of the pain and the stage of the disease should play the most important role in final decisions.

## Figures and Tables

**Figure 1 biotech-10-00011-f001:**
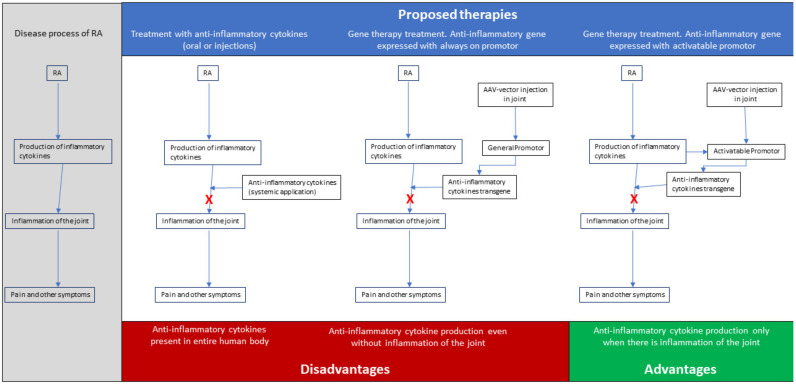
Schematic representation of a proposed treatment of Rheumatoid Arthritis (RA) with gene therapy applications.

## Data Availability

Not applicable.

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
