# Peer review of "Treatment of Rheumatoid Arthritis with Gene Therapy Applications: Biosafety and Bioethical Considerations"

_biotech, 2021, doi:10.3390/biotech10030011_

Round 1

Reviewer 1 Report

The manuscript by Tsitrouli et l. represents an opinion article analyzing the situation with the biosafety and bioethical considerations rising during the potential treatment of rheumatoid arthritis with gene therapy applications. Thus, the authors discuss a number of important and interesting questions, while the manuscript is well written and easy to follow. I have only a couple of suggestions:

- The manuscript misses the conclusions section summarizing the main ideas. Moreover, in my opinion, it would be nice to illustrate it with the scheme.

- I am wondering why the authors do not cite a recent review article (PMID: 31936504) summarizing recent approaches for RA gene therapy

Author Response

The manuscript by Tsitrouli et l. represents an opinion article analyzing the situation with the biosafety and bioethical considerations rising during the potential treatment of rheumatoid arthritis with gene therapy applications. Thus, the authors discuss a number of important and interesting questions, while the manuscript is well written and easy to follow. I have only a couple of suggestions:

We would like to thank the reviewer for the positive disposition towards our article and the useful comments that helped to further improve our original manuscript. Your comments and suggestions have been considered and the relevant necessary changes have been made in the revised text. They are highlighted in red, to facilitate reading.

- The manuscript misses the conclusions section summarizing the main ideas. Moreover, in my opinion, it would be nice to illustrate it with the scheme.

We appreciate the suggestion of the reviewer. We have now added a conclusions section (L214-224), and a scheme illustrating the main idea of this theoretical exercise as Figure 1.

- I am wondering why the authors do not cite a recent review article (PMID: 31936504) summarizing recent approaches for RA gene therapy

Thank you for your recommendation, indeed the mentioned review article is relevant in our discussion, and we now have added it in L58-61.

Reviewer 2 Report

The authors propose the use of gene therapy and gene editing for the treatment of RA.

They state that what they proposed is something new. To my knowledge other papers already described the possibility of using AAV for gene therapy to treat RA, please discuss what is different in your opinion as compared to previous works

- Gene therapy is generally used for treatment of monogenic inherited disorders wherein a mutation in a single gene is accountable for disease onset. RA is a multifactorial chronic autoimmune disease. Please deeply explain why you suggest the use of gene therapy for RA, it will help readers to understand your point. 

- Which gene should be targeted to treat RA in FLS? I think few considerations should be done regarding this issue

- The delivery process (viral and not viral) and the potential for permanent changes to the host cell genome have potential risks that should be discussed in a better way.

- The discussion is focused on Adeno-associated virus (AAV) other delivery systems should be discussed in the manuscript.

- Gene editing technologies is a complex process. Sometime, for the general public, is difficult to understand its mechanisms, possible benefits, and side effects. how would you handle it?

- Conclusions should be added to the manuscript

Author Response

The authors propose the use of gene therapy and gene editing for the treatment of RA.

We would like to thank the reviewer for his comments that helped improve the original manuscript. We have taken into consideration all the suggestions and made necessary changes highlighted in red in the revised text to facilitate reading. Concerning this comment, we would like to highlight that the theoretical exercise we describe in this article only involves gene therapy and not gene editing.

They state that what they proposed is something new. To my knowledge other papers already described the possibility of using AAV for gene therapy to treat RA, please discuss what is different in your opinion as compared to previous works

The reviewer is correct that there is literature describing the possibility of using AAV for gene therapy to treat RA. However, we attempt to make clear the steps that are needed for a successful gene therapy for this disease, with the attached risks and ethical consequences of such a therapy, which to the best of our knowledge is an original approach. The article is not supposed to be read as a new therapy for RA. We tried to make this clearer within the introduction in L 49-52.

- Gene therapy is generally used for treatment of monogenic inherited disorders wherein a mutation in a single gene is accountable for disease onset. RA is a multifactorial chronic autoimmune disease. Please deeply explain why you suggest the use of gene therapy for RA, it will help readers to understand your point. 

The reviewer is correct that most of the applications of gene therapy focus on monogenic diseases. However, most if not all the forms of RA result in the inflammation of the joint and thus share the process of inflammation. By targeting to the inflammatory process, a gene therapy for RA can be successful. Many examples are already described in animal and human studies. We added text to clarify this (L58-61). We also added a reference (PMID: 31936504) in which these examples are mentioned.

- Which gene should be targeted to treat RA in FLS? I think few considerations should be done regarding this issue

We appreciate the reviewer’s comment, however we would like to underline that in our proposed treatment we do not target genes in the FLS, but we put in an anti-inflammatory transgene to suppress inflammation. If we would target genes in the FLS, we probably would never prevent inflammation, because it is not a monogenetic disease, and more than 1 inflammatory cytokines are produced. Instead, we chose a treatment independent of the cause of inflammation, we wanted to overall block inflammation. Please see L 58-61, and L63, we hope we clarified this point.

So it will be very difficult to address this question in the manuscript if it is not a treatment that will be successful.- The delivery process (viral and not viral) and the potential for permanent changes to the host cell genome have potential risks that should be discussed in a better way.

We would like to thank the reviewer for this comment, we tried to further elaborate on it in L70-80.

- The discussion is focused on Adeno-associated virus (AAV) other delivery systems should be discussed in the manuscript.

The reviewer here correctly marks that the discussion is focused on AAV; however, we do appreciate the comment and further added some points mentioning also other delivery systems in L70-80.

- Gene editing technologies is a complex process. Sometime, for the general public, is difficult to understand its mechanisms, possible benefits, and side effects. how would you handle it?

We appreciate your comment, however we would like to highlight that we don’t mention gene editing in our manuscript.

- Conclusions should be added to the manuscript

Thank you for your suggestion. We have now added a conclusions section (L214-224) as well as a scheme illustrating the main idea of this theoretical exercise (Figure 1).

Round 2

Reviewer 2 Report

The authors have done a good job addressing the comments, and I have no further suggestions. The quality of the article has significantly improved and I believe the paper is acceptable for publication in BioTech.